# Removal of Tetracycline by Hydrous Ferric Oxide: Adsorption Kinetics, Isotherms, and Mechanism

**DOI:** 10.3390/ijerph16224580

**Published:** 2019-11-19

**Authors:** Ji Zang, Tiantian Wu, Huihui Song, Nan Zhou, Shisuo Fan, Zhengxin Xie, Jun Tang

**Affiliations:** College of Resource and Environment, Anhui Agricultural University, Key Laboratory of Agri-Food Safety of Anhui Province, Hefei 230036, China; 18895688872@163.com (J.Z.); wu18365280208@163.com (T.W.); songhuihui188@163.com (H.S.); 18726172019@163.com (N.Z.); fanshisuo@126.com (S.F.); xiezx@ahau.edu.cn (Z.X.)

**Keywords:** hydrous ferric oxide, tetracycline, influencing factors, adsorption behavior, mechanism

## Abstract

The removal of tetracycline (TC) from solution is an important environmental issue. Here we prepared an adsorbent hydrous ferric oxide (HFO) by adjusting a FeCl_3_·6H_2_O solution to neutral pH. HFO was characterized by a surface area analyzer, X-ray diffraction (XRD), Fourier transform infrared spectroscopy (FTIR), and X-ray photoelectron spectroscopy (XPS), and was used to remove TC from solution. The influence of pH, solid-to-liquid ratio, ionic type, and strength on TC removal was investigated. Adsorption kinetics and isotherms were also determined. HFO after adsorption of TC was analyzed by FTIR and XPS to investigate the adsorption mechanism. The results showed that the adsorption of TC increased from 88.3% to 95% with increasing pH (3.0–7.0) and then decreased. K^+^ ions had little effect on TC adsorption by HFO. However, Ca^2+^ and Mg^2+^ reduced the adsorption of TC on HFO. When the concentrations of Ca^2+^ and Mg^2+^ were increased, the inhibitory effect was more obvious. Pseudo-second-order kinetics and the Langmuir model fitted the adsorption process well. The maximum adsorption capacity of TC on HFO reached 99.49 mg·g^−1^. The adsorption process was spontaneous, endothermic, and increasingly disordered. Combination analysis with FTIR and XPS showed that the mechanism between TC and HFO involved electrostatic interactions, hydrogen interactions, and complexation. Therefore, the environmental behavior of TC could be affected by HFO.

## 1. Introduction

Antibiotics have received increasing attention due to their various adverse effects in the aquatic environment. Tetracyclines (TCs) represent one of the most widely used antibiotic agents both in veterinary science and aquaculture, with an annual usage of more than 6950 tons in China [1]. TCs in the environment presented at very low concentrations (μg/L to ng/L) in treated waters and higher levels (100–500 mg/L) were detected in effluents from hospital and pharmaceutical manufacturing wastewaters [2,3,4]. TCs can persist in the environment for a long time, and they have high aqueous solubility [5]. Long-term exposure to TCs can cause pathogenic microorganisms to develop antibiotic resistance [6]. Furthermore, resistant genes can spread and evolve in the environment, posing potential threats to ecological environments and human health. Therefore, the removal of antibiotics from the aqueous environment is crucial.

Generally, the techniques used for antibiotic removal from aqueous solution involve adsorption [7,8,9], advanced oxidation [10,11], and biological methods [12]. Due to the advantages of low cost, simple operation, high efficiency, and recycling of the adsorbent, adsorption has become one of the most effective methods for TC removal [13]. Various adsorbent materials have been used for the removal of TCs from aqueous solution, including graphene oxide [14], graphene oxide functionalized magnetic particles [14], chitosan [15], ferric activated sludge-based adsorbent [16], activated carbon [17], and others.

Hydrous ferric oxide (HFO) is a widespread iron oxide present in natural water and soil [18]. HFO is an excellent adsorbent and can remove many pollutants, such as ammonium, phosphate, and fluoride, and can affect the transfer and transformation of pollutants [19,20,21]. The active sites on HFO can bind pollutants via surface complexation or ligand exchange. Pollutant removal from wastewater by HFO or modified HFO has been reported, including phosphorus [20,22], fluoride [23], antimony [24], and heavy metal ions [25], etc. Antibiotic removal by HFO has also been reported, including sulfadimethoxine [8]. Gu et al. [26] reported that HFO can influence the mobility and environmental reactivity of TC in complex environmental matrices. However, the presence of competing cations, the influence of temperature, and potential mechanism should be studied further. In addition, HFO is the most important component of iron plaque (up to 81%–100%) around the roots of hydrophilous plants [27] and plays a key role in the bioaccumulation of various contaminants in wetland plants [28]. Given the effect on uptake and translocation of norfloxacin in rice by iron plaques [29], the adsorption behavior and mechanism of antibiotics by HFO require further investigation to better understand the environmental behavior of antibiotics. 

In this study, HFO was synthesized using FeCl_3_·6H_2_O and was characterized by XRD, FTIR, and XPS. The effect of HFO dosage, pH, ionic types and strength on TC removal by HFO was investigated. The adsorption kinetics and isotherms were also studied. In addition, the adsorption mechanism was determined to better understand the environmental behavior of TC influenced by HFO.

## 2. Materials and Methods

### 2.1. Reagents 

All the chemicals used in this study were of analytical grade. Ferric chloride hexahydrate (FeCl_3_·6H_2_O) and MgCl_2_·6H_2_O were bought from Sinopharm Chemical Reagent Co., Ltd (Shanghai, China). Sodium hydroxide (NaOH), hydrogen chloride (HCl), potassium chloride (KCl), and calcium chloride (CaCl_2_·2H_2_O) were obtained from Xilong Scientific Co., Ltd. (Guangzhou, China). High-performance liquid chromatography grade methyl alcohol and acetonitrile were purchased from Tedia (Fairfield, OH, USA). Tetracycline hydrochloride (purity 98%) was purchased from Shanghai Yuanye Reagent Company (Shanghai, China). Deionized water was obtained from a Milli-Q Plus ultrapure water system (Billerica, MA, USA).

### 2.2. Synthesis of HFO

The synthesis of HFO was based on the study by Gu [26], and the steps were as follows: FeCl_3_·6H_2_O (0.8 mol·L^−1^) was dissolved in deionized water under vigorous magnetic stirring, and then NaOH solution was slowly added into the FeCl_3_·6H_2_O solution until the pH value was 7–8. The formed suspension was continuously stirred for 1 h, and aged at room temperature for 24 h. The precipitated HFO particles were collected by centrifugation and then washed three times with ultrapure Milli-Q water to remove residual ions. Finally, the HFO was freeze-dried, ground in an agate mortar and passed through a 200-mesh sieve. The obtained HFO was collected and stored in a brown glass bottle until used. 

### 2.3. Adsorption Experiments

A standard stock solution of 1000 mg·L^−1^ TC was prepared and stored in the refrigerator at 4 °C. All other experimental concentrations of TC solutions were obtained by diluting the stock solution. Forty milliliters of the desired TC solution was placed in a 100 mL conical flask and oscillated at 298 K and a speed of 150 rpm for batch adsorption experiments. Samples were taken at a given time, centrifuged at 10,000 rpm for 10 min, and then collected for further analysis.

The influence of adsorbent dosage, pH value, and ionic strength on TC adsorption were investigated. Fifty milligrams of HFO was added to 50 mL of TC solution (40 mg·L^−1^) in a conical flask and the initial pH value of the solution was adjusted from 3.0 to 11.0 by adding 0.1 mol·L^−1^ HCl or 0.1 mol·L^−1^ NaOH using a PHS-3C pH meter (Shanghai Electronics Science Instrument Co., Ltd., Shanghai, China). KCl, CaCl_2_, and MgCl_2_ at different concentrations (0.02, 0.10, and 0.50 mol·L^−1^) were selected to analyze the impact of ionic type and strength on adsorption. The experimental time for the effect of pH and ionic types was 7 h. The removal rate is listed in Equation (1).
R = (C_0_ − C_e_)/C_0_ × 100%,(1)
where R is the removal rate, %. C_0_ and C_e_ are the initial and the equilibrium concentration of TC in the solution phase, respectively, mg·L^−1^.

To measure the adsorption kinetics, 0.1 g of HFO was added to the TC solution (40 mg·L^−1^, 40 mL), and the sampling times were set at intervals of 10 min up to 1260 min. The amount of TC adsorbed at t time, q_t_ (mg·L^−1^), was determined according to Equation (2).
q_t_ = (C_0_ − C_e_)V/m,(2)
where q_t_ is the amount adsorbed at equilibrium time, mg·g^−1^; V is the volume of solution, L; m is the mass of adsorbent, g.

To study the adsorption isotherms, different concentrations of TC (10, 20, 30, 40, 50, 60, 70 mg·L^−1^) were placed in a 100 mL conical flask with 0.1 g HFO at 298, 308, and 318 K, respectively, and the agitation time was set at 420 min according to the results of the adsorption kinetics. The amount of TC adsorbed, q_e_ (mg·L^−1^), was calculated by Equation (3).
q_e_ = (C_0_ − C_e_) × V/m.(3)

Pseudo-first-order and pseudo-second-order models were used to fit the adsorption kinetic process, and the Langmuir and Freundlich models were applied to simulate the adsorption isotherm data. 

### 2.4. Measurement of TC

TC was analyzed by ultra-performance liquid chromatography (Waters, Milford, MA, USA) with the column oven temperature maintained at 40 °C, using a BEH C_18_ reversed-phase column (100 mm × 2.1 mm i.d., 1.7 μm). The mobile phase consisted of water containing 1‰ formic acid (A) and acetonitrile (B). The gradient was set as follows: 95.0% A (0 min), 5.0% A (2.5 min), 95.0% A (3.51 min), 95.0% A (5 min) and the flow rate was 0.30 mL·min^−1^. The injection volume was 10 μL, and the detection wavelength was 254 nm. Each experiment was repeated three times, and the average values are presented. 

### 2.5. Characterizations

Determination of pHpzc (point of zero charge): 40 mL of pure water was added to a 50 mL centrifugal tube. The initial pH (pHi) of the solution was adjusted to 2.0–10.0 using 0.1 mol·L^−1^ HCl or 0.1 mol·L^−1^ NaOH. Then, 0.04 g HFO was added to each centrifugal cube. After stirring in a shaker for 24 h, the suspension was centrifuged, and the pH of the supernatant was measured (pHf). The value of pHpzc was obtained when the value of pHi was equal to pHf. 

The minerals in HFO were detected by an X-ray diffractometer with Cu radiation (PANalytical X’Pert PRO X’Celerator). The functional groups in the sample were determined by Fourier transform infrared spectroscopy (FTIR Nicolette is 50, Thermo Fisher Scientific, Waltham, MA, USA) using the KBr pellet technique with 4 cm^−1^ resolution measuring the absorbance from 4000 to 400 cm^−1^. The sample for FTIR analysis was crushed and was passed through a 200 mesh sieve. The valence elements in HFO and HFO-adsorbed TC were analyzed via X-ray photoelectron spectroscopy (XPS, Thermo ESCALAB 250, Thermo-VG Scientific, Waltham, MA, USA). XPS was used to determine the valence state of the chemical elements by analyzing the energy distribution of photoelectrons. The specific surface area, pore volume, and pore size of the HFO were determined using a surface area and porosity analyzer (Micromeritics, Tristar II 3020, Atlanta, GA, USA) at 77 K under N_2_ atmosphere. The particle size of HFO was measured using a BT-9300H laser particle analyzer (Dandong Baite Instrument Co., Ltd., Liaoning, China).

### 2.6. Adsorption Models

Pseudo-first-order and pseudo-second-order models were used to fit the adsorption kinetic process. 

Pseudo-first-order kinetic model [30]:q_t_ = q_e_(1 − exp(−k_1_t)).(4)

Pseudo-second-order kinetic model [31]:q_t_ = (k_2_q_e_^2^t)/(1 + k_2_q_e_t),(5)
where q_t_ (mg·g^−1^) and q_e_(mg·g^−1^) are the amounts of adsorbed at time t and equilibrium time, respectively. The k_1_ (min^−1^) and k_2_ (g·(mg·min^−1^)) are the adsorption rate constant of the pseudo-first-order and pseudo-second-order model, respectively. 

The Langmuir and Freundlich models were applied to fit the adsorption isotherm process. 

Langmuir [32]:q_e_ = (q_m_·K_L_·c_e_)/(1 + K_L_·C_e_).(6)

Freundlich [33]:q_e_ = *K_f_*·C_e_^1/n^,(7)
where q_e_ (mg·g^−1^) is the equilibrium adsorption capacity; q_m_ (mg) is the maximum adsorption capacity. K_L_ is a constant for the Langmuir model. *K_f_* (L·mg^−1^) is a constant for the Freundlich model, and 1/n is the adsorption affinity constant.

To better understand the effect of temperature on the adsorption of TC on the HFO composite, the thermodynamic parameters of the adsorption process, such as a change in standard free energy (ΔG), enthalpy (ΔH), and entropy (ΔS), were calculated using Equations (9)–(11) [34,35].
(8)$$K_{d}=\frac{q_{e}}{C_{e}},$$(9)$$\Delta G=-\mathrm{RTln}(K_{d}),$$(10)$${\mathrm{lnK}}_{d}=\frac{\Delta S}{R}-\frac{\Delta H}{R}\frac{1}{T},$$
where K_d_ is the apparent equilibrium constant; R is the ideal gas constant (8.314 J mol^−1^ K^−1^), and T is Kelvin temperature (K).

According to the method suggested by Khan and Singh [34], the sorption distribution coefficient K_d_ for the sorption reaction was determined from the slope of the plot ln(q_e_/C_e_) against C_e_ at various temperatures and extrapolating to zero C_e_. The values of ΔH and ΔS can be obtained from the slope and intercept of a plot of ln(K_d_)against 1/T.

## 3. Results and Discussion

### 3.1. Characterization of HFO

The specific surface area, pore volume, and pore size of the HFO were 226.796 m^2^·g^−1^, 0.167 cm^3^·g^−1^, and 3.495 nm, respectively. The D10, D50, and D90 of HFO particles were 5.279, 27.70, and 65.89 μm, respectively. Thus, the pore structure of HFO was developed, and the particles of HFO was relatively large, which was beneficial for TC adsorption.

The XRD of HFO is presented in Figure 1a. The XRD pattern indicated that HFO had poor crystallinity. HFO showed two broad peaks at 36.4° and 64.2°, which matched with the poorly ordered ferrihydrite mineral. Hofmann et al. [36] found that these peaks were similar to ferrihydrite.

The FTIR spectra of HFO, TC, and TC adsorbed onto HFO are displayed in Figure 1b, and specific functional groups are listed in Table 1. The main functional groups of HFO included –OH (3405 cm^−1^, 1629 cm^−1^), CH_3_/COO (1477 cm^−1^, 1348 cm^−1^), and Fe-O (444 cm^−1^). After TC was adsorbed on HFO, some new peaks appeared, involving amino groups (1535 cm^−1^) and C-OH stretching (1224 cm^−1^), suggesting that TC had been adsorbed on the surface of HFO.

In addition, following TC adsorption, the peaks at 1629 cm^−1^, 1477 cm^−1^, and 444 cm^−1^ shifted to 1617 cm^−1^, 1458 cm^−1^, and 436 cm^−1^, respectively, indicating that Fe-O, CH_3_/COO and -OH in HFO participated in the removal of TC in solution. The main mechanism involved electrostatic interactions and complexation. Mudunkotuwa et al. [37] found that the ATR-FTIR spectra of α-Fe_2_O_3_-adsorbed humic acid showed two new bands at 1348 and 1470 cm^−1^ during the initial time points, which corresponded to carboxylate groups strongly adsorbed to the surface iron atoms.

The XPS spectra of HFO and HFO-adsorbed TC are presented in Figure 2. The XPS-peak-differentiating analysis of O 1s is shown in Figure 2a. The O 1s XPS spectrum of TC can be separated into three peaks at 531.30, 532.45, and 533.24 eV, corresponding to C=O, -OH, and C-O-C, respectively [13]. The O 1s XPS spectrum of HFO can be separated into three peaks at 529.75, 531.20, and 531.58 eV, corresponding to Fe-O, Fe-OH, and C=O/-OH, respectively [8]. Following TC adsorption, the O 1s can be separated into three peaks at 529.84, 531.23, and 532.07 eV, which were ascribed to Fe-O, Fe-OH, and C=O/-OH, respectively. Therefore, the peak position and peak area of oxygen-containing groups in HFO obviously changed after TC adsorption. Thus, Fe-O, Fe-OH, and C=O/-OH participated in the adsorption of TC, which mainly involved electrostatic interactions and surface complexation.

The XPS-peak-differentiating analysis of N 1s is displayed in Figure 2b. The N 1s XPS spectrum of TC can be separated into two peaks at 399.65 and 401.92 eV, corresponding to NH/NH_2_ and C-N-H, respectively [38]. Following TC adsorption, the N 1s was separated into one peak at 399.67 eV, which was ascribed to NH/NH_2_. Therefore, the peak position and peak area of nitrogen-containing groups in HFO obviously changed after TC adsorption. Thus, nitrogen-containing groups participated in the adsorption of TC, which mainly involved electrostatic and hydrogen interactions.

The XPS-peak-differentiating analysis of Fe 2p is presented in Figure 2c. The Fe 2p XPS spectrum of HFO was separated into four peaks at 710.50, 712.23, 718.51, and 724.63 eV, corresponding to Fe 2p_3/2_, Fe 2p_3/2_, Fe 2p_1/2_, and Fe 2p_1/2_, respectively. The Fe species in HFO refers to Fe_2_O_3_, FeO, FeOOH, which was consistent with the analysis of XRD. Following TC adsorption, the Fe 2p spectrum was separated into four peaks at 710.44, 712.16, 718.50, and 724.58 eV, which were ascribed to Fe 2p_3/2_, Fe 2p_3/2_, Fe 2p_1/2_, and Fe 2p_1/2_, respectively. Therefore, the peak position and peak area of iron-containing groups in HFO obviously changed after TC adsorption. The iron-containing groups participated in the adsorption of TC, which mainly involved complexation.

As shown in Figure 3, the pHpzc value of HFO was 7.87. When the pH of the solution was lower than 7.87, the surface of HFO had a positive charge. The surface of HFO had a negative charge when the pH of the solution was greater than 7.87. Okazaki et al. [39] found that the isoelectric point of HFO was 7.5. Kosmulski et al. [40] found that the isoelectric point of HFO was 7.2.

### 3.2. Effect of HFO Dosage, pH, Ionic Types and Strength

The effect of HFO dosage on TC removal is shown in Figure 4a. When the dosage of HFO was increased, the removal rate of TC quickly increased and then tended to stabilize. When the HFO dosage increased from 0.5 to 1.0 g·L^−1^, the removal rate of TC increased from 67.17% to 95.08%. When the HFO dosage was larger than 1.0 g·L^−1^, the removal rate tended to stabilize, and the adsorption capacity reached 38.18 mg·g^−1^. Thus, the optimum HFO dosage was chosen as 1.0 g·L^−1^.

Solution pH was a key factor in affecting TC adsorption. Solution pH can affect the surface charge and ionization degree of HFO, the form, and the ionization degree of TC, consequently influencing the adsorption of TC on HFO. Tetracycline (symbolized as H_2_TC) is an amphoteric molecule with multiple ionizable functional groups and may exist as a cations (H_3_TC^+^, pH < 3.3), zwitterions (H_2_TC^0^, 3.3 < pH < 7.7), or negatively charged ions (HTC^−^, 7.7 < pH < 9.7; TC_2_^−^, pH > 9.7) at different pH values [41]. The effect of pH on TC removal by HFO is shown in Figure 4b. The adsorption of TC on HFO increased from 88.3% to 95% with increasing pH (3–7) and then decreased. When the solution pH was 3.0, the removal rate was greater than 90%, despite the presence of electrostatic repulsion, indicating other adsorption mechanisms also played a role. The removal rate of TC increased in the pH range of 3.0 to 5.0, due to the weakening of electrostatic repulsion. The removal rate slightly decreased at pH 5.0–9.0, although electrostatic repulsion was present, and it had little effect, suggesting other mechanisms played larger roles. However, the removal rate of TC significantly decreased when the pH value was greater than 9.0, indicating that electrostatic repulsion between TC and HFO dominated. When TC was removed by other adsorbents, such as iron hydrous oxides [26], iron-montmorillonite [42], montmorillonite [43], porous synthetic resins [44], and carbon nanotubes [45], it was found that at increased pH, TC removal first increased and then decreased, which was attributed to complexation interactions, electrostatic interactions, ion exchange, cation–π bonding, and π–π EDA interactions.

Wastewater may contain high concentrations of salt ions, and the salt in water may affect the adsorption of TC by HFO. In this study, the effects of K^+^, Ca^2+^, and Mg^2+^ at different concentrations on TC adsorption by HFO were investigated, and the results are shown in Figure 5. It can be seen that K^+^ has little influence on TC removal by HFO. However, Ca^2+^ and Mg^2+^ had an obvious influence on TC removal. As the concentration of Ca^2+^ and Mg^2+^ increased, the removal rates of TC significantly decreased. When the concentration of K^+^ increased from 0 to 0.5 mol·L^−1^, the removal rate of TC was almost unchanged. However, when the concentration increased from 0 to 0.5 mol·L^−1^, the removal rate of TC decreased from 96.50% to 17.18% for Ca^2+^and from 96.50% to 27.13% for Mg^2+^, respectively. Thus, Ca^2+^and Mg^2+^ in the solution reduced the removal of TC by HFO.

Ca^2+^ and Mg^2+^may affect TC removal by competing with TC for adsorption sites on HFO; thus influencing complexation interactions, especially outer-sphere complexation, affecting the ion exchange mechanism, weakening the electrostatic interaction, and Ca^2+^ and Mg^2+^ could form a complex with TC, further affecting the adsorption of TC on HFO. Parolo et al. [46] reported that in the absence of Ca^2+^, TC adsorption was high at low pH and decreased as the pH increased. Cation exchange was the prevailing process at pH < 5.0; thus, TC adsorption decreased due to increasing total Ca^2+^ concentration. In contrast, Ca-bridging was the prevailing process at pH > 5.0; thus, TC adsorption increased with increasing Ca^2+^ concentration. Zhao et al. [47] investigated the adsorption of TC on kaolinite and found that Ca^2+^and Mg^2+^ inhibited the adsorption of TC on kaolinite due to competition between Ca, Mg, and TC. Li et al. [48] investigated TC adsorption by activated sludge and found that TC competed with Ca^2+^and Mg^2+^ ions at the adsorption sites in sludge.

### 3.3. Adsorption Kinetics

The adsorption kinetics and fitting curves of TC on HFO are shown in Figure 6. As the adsorption time increased, the amount of TC adsorbed by HFO initially increased and then tended to stabilize. Rapid adsorption of TC on HFO can be ascribed to adequate adsorption sites. The removal rate was greater than 90% within 200 min.

The fitting parameters of adsorption kinetics are presented in Table 2. It can be seen that compared with the pseudo-first-order kinetic model, the pseudo-second-order model better described the adsorption process and showed a higher correlation coefficient (>0.99). Thus, chemical interactions played a major role in the adsorption process [49].

### 3.4. Adsorption Isotherms

The adsorption isotherm and fitting curve of TC on HFO are shown in Figure 7. The influence of temperature on the adsorption isotherm is also presented. With increased equilibrium concentration, the adsorption capacity increased and then tended to stabilize. Increased temperature was beneficial for the removal of TC by HFO. Increasing temperature can affect adsorption by changing the sorption capacity and also influencing the molecules diffusion rate, thus increasing or decreasing the time for adsorption equilibrium and then enhanced the adsorption of TC on HFO.

The fitting parameters of the adsorption isotherms are listed in Table 3. Compared with the Freundlich model, the Langmuir model fitted the adsorption isotherm process better. The maximum adsorption capacity of TC on HFO reached 99.49 mg·g^−1^ at 318 K. Compared with other iron oxide-based adsorbents [14,42,50,51,52,53], the maximum adsorption capacity of TC on HFO was high, indicating that HFO can effectively adsorb TC and further influence the behavior of TC in the aquatic environment. Adsorption proceeded easily when the 1/n was in the range of 0.1–0.5 [54]. However, little adsorption occurred when 1/n was larger than 2. In this study, the value of 1/n was less than 0.5 under the three temperatures, indicating that TC was easily adsorbed onto HFO.

### 3.5. Adsorption Thermodynamics

A plot of ln(K_d_) against 1/T is shown in Figure 8. The thermodynamic fitting parameters of TC adsorption on HFO are listed in Table 4. The ΔG was −8.7403, −10.1915, and −12.0669 kJ mol^−1^ at 298, 308, and 318 K, respectively. Therefore, the adsorption process was spontaneous. When the temperature increased, spontaneous adsorption became stronger. The ΔH > 0 indicated that the adsorption process was endothermic. ΔS > 0 showed that the disorder of the adsorption system increased. ΔH was 10.1431 kJ mol^−1^, which was less than 80 kJ mol^−1^ [55], indicating that physical binding also plays an important role during the adsorption process.

### 3.6. Adsorption Mechanism

The mechanism between HFO/modified HFO and pollutant included surface complexation, ion exchange, and electrostatic interactions [26,43,45,53,56]. In this study, the adsorption mechanism of TC on HFO was examined by combining the influencing factors, adsorption kinetics, adsorption isotherm, and XPS and FTIR characterization of HFO after adsorption. The effect of pH indicated that electrostatic interactions affected the adsorption of TC on HFO. The removal rate of TC decreased slightly at pH< 5.0 and decreased significantly at pH >9.0. The effect of ionic types and strengths showed that K^+^ had little influence on TC removal, suggesting that the complexation between HFO and TC may be an inner-sphere complexation and not an outer-sphere complexation. The combination of TC and HFO was tight. The results of adsorption kinetics illustrated that chemisorption played a major role in the adsorption process.

According to the results of the adsorption isotherm, high temperature was beneficial for the adsorption process, and adsorption occurred spontaneously. The FTIR analysis showed that Fe-O and -OH were involved in the removal of TC, which included electrostatic interactions and complexation. XPS analysis demonstrated that oxygen-containing groups participated in electrostatic interactions and complexation, nitrogen-containing groups were involved in electrostatic interactions and hydrogen bonding, and iron-containing groups were involved in complexation. Thus, the main mechanism of TC adsorption on HFO included electrostatic interactions, hydrogen bonding, and complexation.

## 4. Conclusions

The adsorption behavior of TC onto HFO could be affected by solution pH, ionic types, and temperature. Pseudo-second-order adsorption kinetics and the Langmuir adsorption isotherm fitted the adsorption process well. Electrostatic interactions, hydrogen bonding, and complexation were the main adsorption mechanism between TC and HFO. Therefore, the environmental behavior of TC could be affected by HFO, and the transformation of TC affected by HFO in the aquatic environment should be studied in the future.

## Figures and Tables

**Figure 1 ijerph-16-04580-f001:**
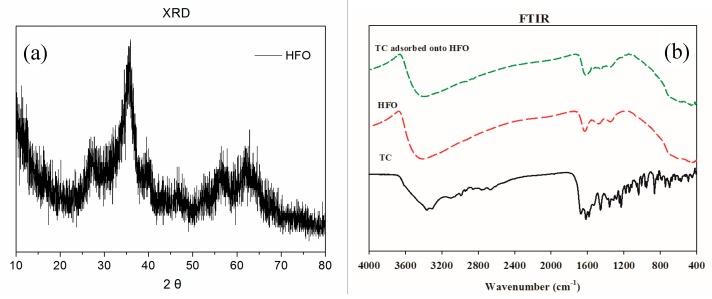
XRD (**a**) and FTIR (**b**) spectra of hydrous ferric oxide (HFO).

**Figure 2 ijerph-16-04580-f002:**
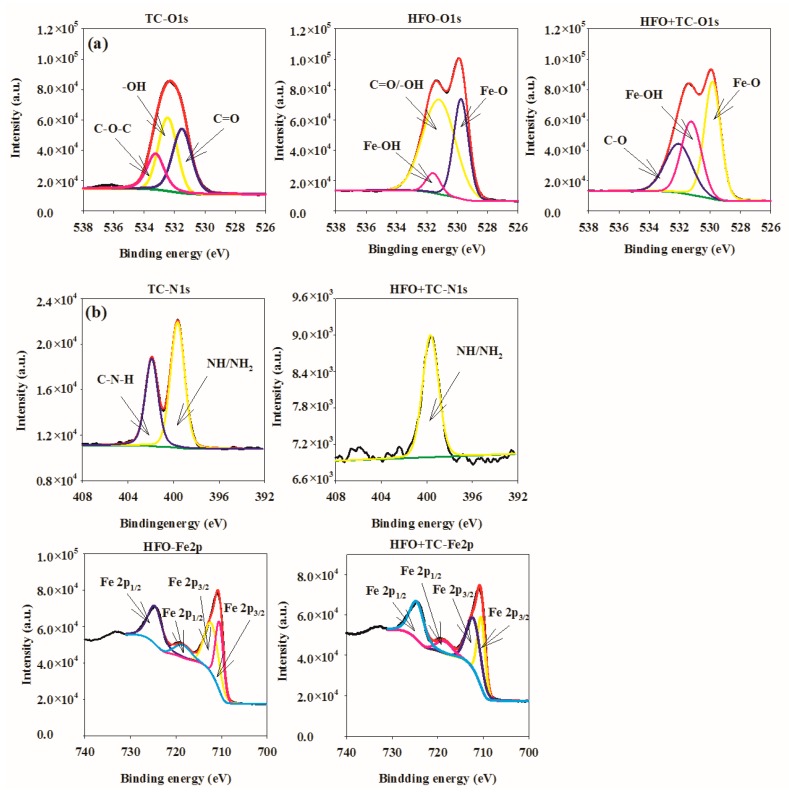
XPS spectral of HFO and HFO adsorbed tetracycline (TC).

**Figure 3 ijerph-16-04580-f003:**
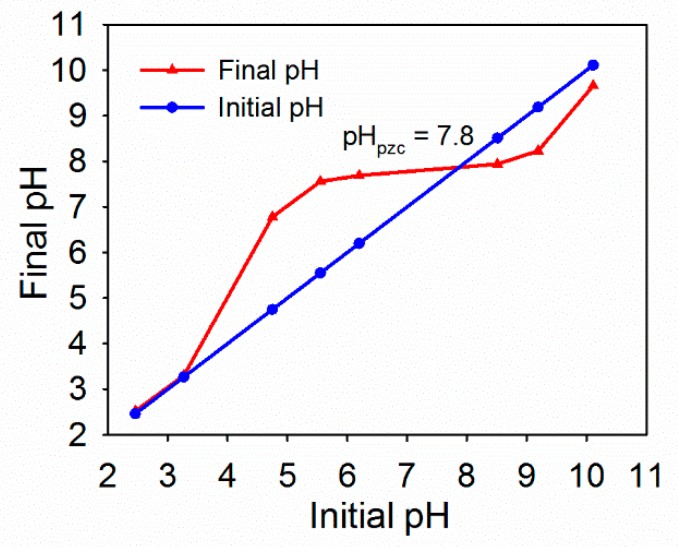
Point of zero charge (pHpzc) of HFO.

**Figure 4 ijerph-16-04580-f004:**
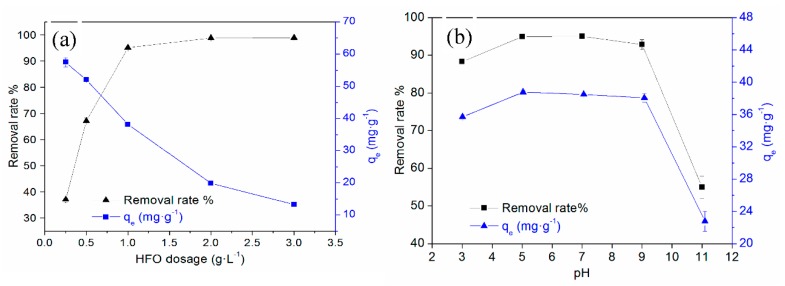
Effect of HFO dosage (**a**) and pH (**b**) on TC removal.

**Figure 5 ijerph-16-04580-f005:**
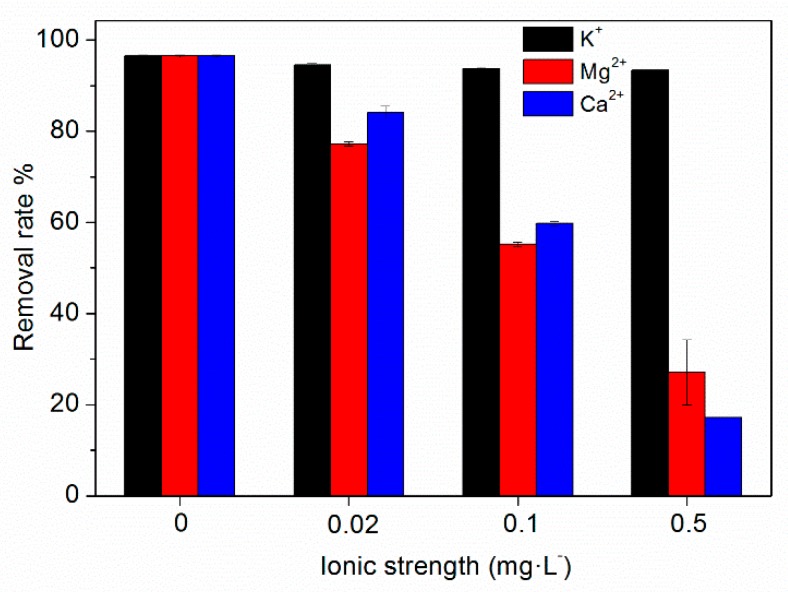
Effect of ionic types and strength on TC removal by HFO.

**Figure 6 ijerph-16-04580-f006:**
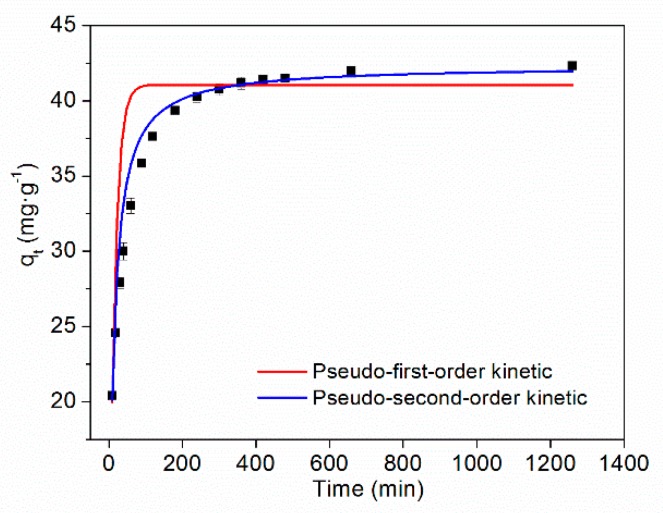
Adsorption kinetics of TC on HFO.

**Figure 7 ijerph-16-04580-f007:**
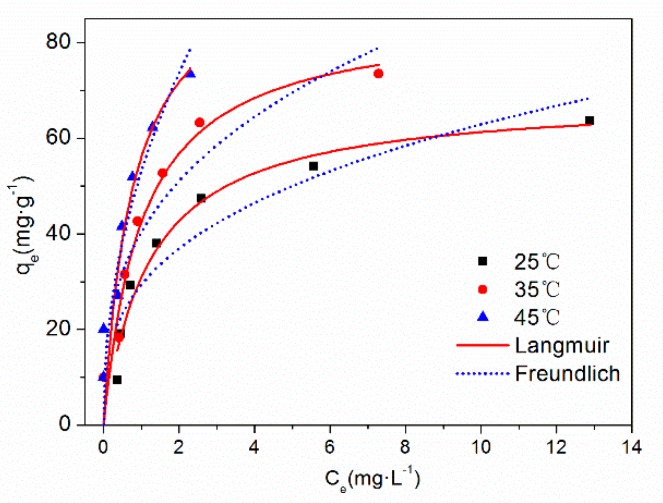
Adsorption isotherm of TC on HFO at different temperatures.

**Figure 8 ijerph-16-04580-f008:**
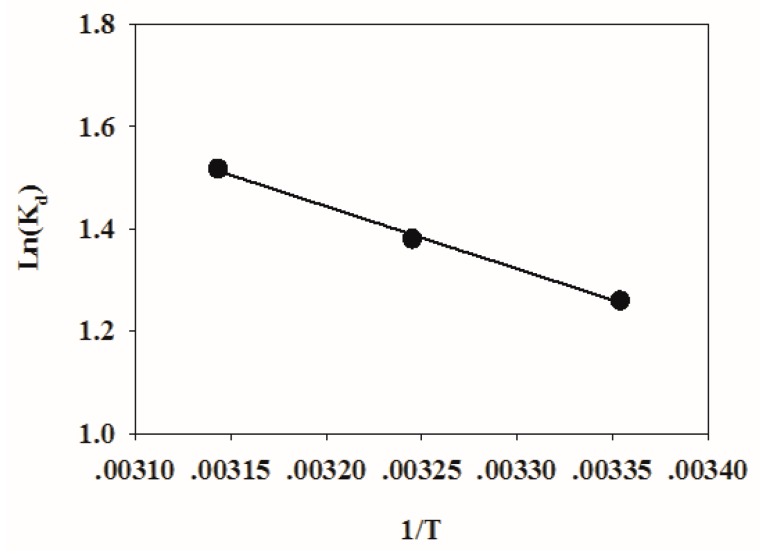
Plotting of 1/T to ln(K_d_).

**Table 1 ijerph-16-04580-t001:** Functional groups of hydrous ferric oxide (HFO) and HFO + tetracycline (TC).

HFO		HFO + TC	
Wavenumber (cm^−1^)	Assignment	Wavenumber (cm^−1^)	Assignment
3405	stretching vibration mode of lattice water	3387	stretching vibration mode of lattice water
1629	-OH	1618	carbonyl groups
		1535	amino groups
1477	CH_3_/COO	1458	CH_3_/COO
1343	CH_3_/COO	1383	CH_3_/COO
		1224	C-OH stretching CO and OH groups
444	Fe-O	436	Fe-O

**Table 2 ijerph-16-04580-t002:** Fitting parameters of adsorption kinetic.

Pseudo-First Order Equation			Pseudo-Second-Order Equation		
k_1_ (min^−1^)	q_e_ (mg·g^−1^)	R^2^	k_2_ (g·(mg·min)^−1^)	q_e_ (mg·g^−1^)	R^2^
0.06662	41.06197	0.96382	0.00214	42.3403	0.99479

**Table 3 ijerph-16-04580-t003:** Parameters of adsorption isotherm.

T(K)	Langmuir Model			Freundlich Model		
	K_L_(L·mg^−1^)	q_m_(mg·g^−1^)	R^2^	K_F_(mg·g^−1^)(L·μg^−1^)^1/n^	n	R^2^
298	0.82377	70.58543	0.97367	25.89622	0.3681	0.86966
308	0.99445	87.58641	0.95776	41.95165	0.5040	0.93145
318	1.29393	99.48959	0.94697	53.24752	0.4331	0.88921

**Table 4 ijerph-16-04580-t004:** Parameters of thermodynamic models for the adsorption of TC onto HFO.

Temperature (K)	ΔG (kJ mol^−1^)	ΔH (kJ mol^−1^)	ΔS (J mol^−1^ K^−1^)
298	−8.7403	10.1431	44.4716
308	−10.1915		
318	−12.0669		

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
