# Peer review of "Removal of Tetracycline by Hydrous Ferric Oxide: Adsorption Kinetics, Isotherms, and Mechanism"

_ijerph, 2019, doi:10.3390/ijerph16224580_

Round 1

Reviewer 1 Report

The manuscript titled “Removal of tetracycline by hydrous ferric oxide: adsorption kinetics, isotherms and mechanism” presents a very complete study, well performed and discussed. However, I have some observations that need to be attended.

The section related to material characterization is the weakest of the manuscript. XRD pattern is very noisy and with low intensity, authors did not add the index for the reflections at 36° and 64°, and they do not discuss the resulting crystallinity.

What are the particle size and the specific surface area of the material?. Those parameters are important for adsorbent materials.

Increase the quality of figure 1 b), titles in plot and numbers are very small. Figure caption needs to be modified; it said “FTIR spectra of HFO” but it also includes TC and modified HFO spectra. In the FTIR discussion, both HFO and HFO-TC spectra are very similar. Following the lines in the figure, there is no shift as the authors claim.

What is the resolution of the FTIR spectra? because if it is about 1 cm-1 or higher, the authors can round the figures in table 1.

Add experimental details about XPS, FTIR, and XRD characterization, conditions, parameters and equipment. In my opinion, the deconvolution of XPS spectra is not well performed mainly for the deconvolution of O1s, since there is a big difference between FWHM values of the peaks in each particular spectrum.

In figure 2, the spectra titles should be O1s, N1s and Fe2p respectively.

On page 5 line 184, authors mentioned a contribution C-C, but they didn’t show the C1s spectrum.

During the adsorption experiments at different pH values, the authors explain an increase in the adsorption between ph 3 to 9. Can the authors add the standard deviation data for this experiment? From the plot, it seems the removal rate shifted around the same value in that ph range.

Add definition or calculation of removal rate.

Can the authors discuss their findings from the adsorption isotherms with other iron oxide-based adsorbents reported in the literature 

Author Response

Reply to the Comments of Reviewer 1

Comment 1: The section related to material characterization is the weakest of the manuscript. XRD pattern is very noisy and with low intensity, authors did not add the index for the reflections at 36° and 64°, and they do not discuss the resulting crystallinity.

Response 1:

Thanks very much for your comment.

The XRD pattern of HFO was characterized once again and was added in the manuscript. The content referring to 36° and 64° had been added in the revised manuscript. Meanwhile, the surface area, pore volume, pore size and particle size was measured and the result was added in the revised manuscript.

“The specific surface area, pore volume and pore size of the HFO was 226.796 m2·g-1, 0.167 cm3·g-1 and 3.495 nm, respectively. The D10, D50 and D90 of HFO particle was 5.279 μm, 27.70 μm and 65.89 μm, respectively. Thus, pore structure of HFO was developed and the particles of HFO was relative large, which was beneficial for TC adsorption.”

“The XRD pattern indicated that HFO had poor crystallinity. HFO showed two broad peaks at 36.4º and 64.2º, which matched with the poorly ordered ferrihydrite mineral. Hofmann et al. [36] found that these peaks were similar with ferrihydrite. ”

Hofmann, A.; Pelletier, M.; Michot, L.; Stradner, A.; Schurtenberger, P.; Kretzschmar, R. Characterization of the pores in hydrous ferric oxide aggregates formed by freezing and thawing. J. Colloid Interface Sci. 2004, 271, 163-173.

Comment 2: What are the particle size and the specific surface area of the material?. Those parameters are important for adsorbent materials.

Response 2:

Thanks very much for your comment.

The particle size and the specific surface area of the HFO was measured and was added in the revised manuscript.

“The specific surface area, pore volume and pore size of the HFO was determined using a surface area and porosity analyzer (Micromeritics, Tristar II 3020, USA) at 77 K under N2 atmosphere. The particle size of HFO was measured using a BT-9300H laser particle analyzer (Dandong Baite instrument Co., Ltd, China).”

“The specific surface area, pore volume and pore size of the HFO was 226.796 m2·g-1, 0.167 cm3·g-1 and 3.495 nm, respectively. The D10, D50 and D90 of HFO particle was 5.279 μm, 27.70 μm and 65.89 μm, respectively. Thus, pore structure of HFO was developed and the particles of HFO was relative large, which was beneficial for TC adsorption “

Comment 3: Increase the quality of figure 1 b), titles in plot and numbers are very small. Figure caption needs to be modified; it said “FTIR spectra of HFO” but it also includes TC and modified HFO spectra. In the FTIR discussion, both HFO and HFO-TC spectra are very similar. Following the lines in the figure, there is no shift as the authors claim.

Response 3:

Thanks very much for your comment.

The quality of the figure 1b) had been increased in the revised manuscript.

Meanwhile, the titles and caption of the figure also changed.

Some functional groups shifted after TC adsorption. Specifically, In addition, following TC adsorption, the peaks at 1629 cm-1 and 444 cm-1 shifted to 1617 cm-1 and 436 cm-1, respectively, indicating that Fe-O and-OH in HFO participated in the removal of TC in solution.

Comment 4: What is the resolution of the FTIR spectra? because if it is about 1 cm-1 or higher, the authors can round the figures in table 1.

Response 4:

Thanks very much for your comment.

The resolution of the FTIR spectra had been added in the revised manuscript.

The figures had been rounded in the table 1.

The functional groups in sample were determined by Fourier Transform Infrared Spectroscopy using the KBr pellet technique with 4 cm-1 resolution measuring the absorbance from 4000 to 400 cm-1. The sample for FTIR analysis was crushed and was passed through a 200 mesh sieve.

Comment 5Add experimental details about XPS, FTIR, and XRD characterization, conditions, parameters and equipment. In my opinion, the deconvolution of XPS spectra is not well performed mainly for the deconvolution of O1s, since there is a big difference between FWHM values of the peaks in each particular spectrum.

Response 5:

Thanks very much for your comment.

The details of the XPS, XRD and FTIR characterization had been added in the revised manuscript.

The deconvolution of XPS spectra was performed by XPSPEAK41 software. When the FWHM values were adjusted to small, the fitting results were not good.

“The minerals in HFO were detected by X-ray diffractometer with Cu radiation (PANalytical X'Pert PRO X'Celerator). The functional groups in sample were determined by Fourier Transform Infrared Spectroscopy (FTIR Nicolette is50, Thermo Fisher Scientific, NY, USA) using the KBr pellet technique with 4 cm-1 resolution measuring the absorbance from 4000 to 400 cm-1. The sample for FTIR analysis was crushed and was passed through a 200 mesh sieve. The valence elements in HFO and HFO-adsorbed TC were analyzed by via X-ray photoelectron pectroscopy (XPS, Thermo ESCALAB 250, Thermo-VG Scientific, USA). XPS was used to determine the valence state of the chemical elementsby analyzing the energy distribution of photoelectrons. The specific surface area, pore volume and pore size of the HFO was determined using a surface area and porosity analyzer (Micromeritics, Tristar II 3020, USA) at 77 K under N2 atmosphere.”

Comment 6: In figure 2, the spectra titles should be O1s, N1s and Fe2p respectively.

Response 6:

Thanks very much for your comment.

The spectra titles had been changed in the revised manuscript.

Some content also had been revised in the revised manuscript.

Comment 7: On page 5 line 184, authors mentioned a contribution C-C, but they didn’t show the C1s spectrum.

Response 7:

Thanks very much for your comment.

The sentence had been added in the revised manuscript.

“Following TC adsorption, the O 1s can be separated into three peaks at 529.84, 531.23, and 532.07eV, which were ascribed to Fe-O, Fe-OH, and C=O/-OH, respectively. Therefore, the peak position and peak area of oxygen-containing groups in HFO obviously changed after TC adsorption. Thus, Fe-O, Fe-OH, and C=O/-OH participated in the adsorption of TC, which mainly involved electrostatic interactions and surface complexation. ”

Comment 8: During the adsorption experiments at different pH values, the authors explain an increase in the adsorption between ph 3 to 9. Can the authors add the standard deviation data for this experiment? From the plot, it seems the removal rate shifted around the same value in that ph range.

Response 8:

Thanks very much for your comment.

The standard deviation data had been added in the revised figures.

The removal rate slightly decreased at pH 5.0-9.0, although electrostatic repulsion was present and it had little effect, suggesting other mechanisms played larger roles. According to the analysis of FTIR and XPS, surface complexaiton and hydrogen bonding can played an important role for TC removal by HFO.

Comment 9: Add definition or calculation of removal rate.

Response 9:

Thanks very much for your comment.

The removal rate had been defined in the revised manuscript.

R= (C0-Ce)/C0×100% (1)

Where R is removal rate, %. C0 and Ce are the initial and the equilibrium concentration of TC in the solution phase, respectively, mg·L-1.

Comment 10:

Can the authors discuss their findings from the adsorption isotherms with other iron oxide-based adsorbents reported in the literature 

Response 10:

Thanks very much for your comment.

Some discussion had been added in the revised manuscript.

“Compared with other iron oxide-based adsorbents [11, 39, 47-50], the maximum adsorption capacity of TC on HFO was high, indicating that HFO can effective adsorb TC and further influence the behavior of TC in aquatic environment.”

[11] Lin, Y.; Xu, S.; Li, J. Fast and highly efficient tetracyclines removal from environmental waters by graphene oxide functionalized magnetic particles. Chem. Eng. J. 2013, 225, 679-685.

[39] Wu, H.; Xie, H.; He, G.; Guan, Y.; Zhang, Y. Effects of the pH and anions on the adsorption of tetracycline on iron-montmorillonite. Appl. Clay Sci.2016, 119, 161-169.

[47] Peng, L., Ren, Y., Gu, J., Qin, P., Zeng, Q., Shao, J.,Lei, M., Chai, L. Iron improving bio-char derived from microalgae on removal of tetracycline from aqueous system. Environ. Sci. Pollut. Res. 2014, 21, 7631-7640.

[48] Zhang, Z., Lan, H., Liu, H., Li, H., Qu, J. Iron-incorporated mesoporous silica for enhanced adsorption of tetracycline in aqueous solution. RSC Adv. 2015, 5, 42407-42413.

[49] Zhou, Y., Liu, X., Xiang, Y., Wang, P., Zhang, J., Zhang, F., Wei, J., Luo, L., Lei, M., Tang, L. Modification of biochar derived from sawdust and its application in removal of tetracycline and copper from aqueous solution: adsorption mechanism and modelling. Bioresou. Technol. 2017, 245, 266-273.

[50] Tanis, E.; Hanna, K.; Emmanuel, E. Experimental and modeling studies of sorption of tetracycline onto iron oxides-coated quartz. Colloids Surf. A 2008, 327, 57-63.

Reviewer 2 Report

see attached file

Author Response

Reply to the Comments of Reviewer 2

Comment 1:

Line 88-89: Please clarify the meaning of “Samples were taken at given time intervals, centrifuged at 10,000 rpm for 10 min and then collected for further analysis”.
Response 1:

Thanks very much for your comment.

The details of the experimental time had been added in the effect of pH, adsorption kinetics and adsorption isotherm.

“The experimental time for the effect of pH and ionic types was 7h.”

“To measure the adsorption kinetics, 0.1 g of HFO was added to the TC solution (40 mg·L-1, 40 mL) and the sampling times were set at intervals of 10 min up to 1,260 min.”

“To study the adsorption isotherms, different concentrations of TC (10, 20, 30, 40, 50, 60, 70mg·L-1) were placed in a 100 mL conical flask with 0.1g HFO at 298, 308, and 318K, respectively, and the agitation time was set at 420 min according to the results of the adsorption kinetics.”

Comment 2:  

Line 96-98: Please clearly state the volume of solution used for adsorption kinetics.

Response 2:

Thanks very much for your comment.

The volume of solution had been added in the revised manuscript.

“To measure the adsorption kinetics, 0.1 g of HFO was added to the TC solution (40 mg·L-1, 40 mL) and the sampling times were set at intervals of 10 min up to 1,260 min”

Comment 3: Line 121-125: Please add the amount of adsorbent used for determination of point of zero charge.

Response 3:

Thanks very much for your comment.

The amount of adsorbent for determination of point of zero charge had been added in the revised manuscript.

“Then, 0.04 g HFO was added to each centrifugal cube”

Comment 4: Line 126-127, 166 and 178: Please clarify the meaning of “HFO-adsorbed TC”?

Response 4:

Thanks very much for your comment.

The HFO-adsorbed TC means after TC adsorption, the TC remains on the surface of HFO. In order to clear expression, the “HFO-adsorbed TC” had been changed to “TC adsorbed onto HFO”.

Comment 5: Line 150: What was the value of ideal gas constant used for calculations of thermodynamic parameters?

Response 5:

Thanks very much for your comment.

The ideal gas constant (R) is the ideal gas constant (8.314 J mol−1 K−1) and the related content had been changed in the revised manuscript.

Commment 6: Figure 4(b), when the solution pH is between 7.7 and 9.7, the electrostatic repulsion between TC and HFO dominates as PZC of HFO is 7.8. Due to which the TC removal rate should decrease significantly, but the figure shows a slight decrease. Additionally removal rate of TC shall decrease as well when the difference between solution pH and PZC become smaller. Please critically discuss this issue.

Response 6:

Thanks very much for your comment.

In this study, the removal rate slightly decreased at pH 5.0-9.0, although electrostatic repulsion was present and it had little effect, suggesting other mechanisms played larger roles.

When the solution pH is between 7.7 and 9.7, the electrostatic repulsion between TC and HFO dominates as PZC of HFO is 7.8. However, the removal rate did not significantly decreased since other mechanisms may exist. According to the analysis of FTIR and XPS, surface complexaiton and hydrogen bonding can played an important role for TC removal by HFO.

Some content had been added in the revised manuscript.

Comment 7: Line 234: Following above point, what are zwitterions present in TC at 7.7 <pH<9.7?

Response 7:

Thanks very much for your comment.

TC presented in the form of “HTC“ in the range of 7.7<pH<9.7”.

The content was displayed in the revised manuscript.

Comment 8: Line 253-255: It is stated in manuscript “When the concentration of K+ increased from 0 to 0.5 mol·L-1, the removal rate of TC was almost unchanged. However, when the K+ concentration increased from 0 to 0.5 mol·L-1, the removal rate of TC decreased from 96.50% to 17.18% for Ca2+ and from 96.50% to 27.13% for Mg2+, respectively.” How did you draw that conclusion, while Figure 5 shows no effect of K+ on TC adsorption?

Response 8:

Thanks very much for your comment.

The description in the manuscript was wrong and the related content had been revised in the revised manuscript.

“However, when the concentration increased from 0 to 0.5 mol·L-1, the removal rate of TC decreased from 96.50% to 17.18% for Ca2+and from 96.50% to 27.13% for Mg2+, respectively. Thus, Ca2+and Mg2+ in the solution reduced the removal of TC by HFO.”

Comment 9: Line 283-284: The results of Parolo et al. [43] showed that “TC adsorption increased with increasing Ca2+ concentration”. However, the results of present study shows competitive adsorption of Ca2+ with TC on HFO and TC adsorption decreased with increasing Ca2+ concentration. What are the key arguments for your claim

Response 9:

Thanks very much for your comment.

In the Parolo’s study, the adsorbent is montmorillonite. The montmorillonite contains much minerals compoistion. The minerals in montmorillonite can exchange with TC during the adsorption process. When the Ca presentd in the solution, Ca-bridging was the prevailing processat pH>5.0; thus, TC adsorption increased with increasing Ca2+ concentration.  Ca, montmorillonite and TC can form ternary complex and then enhanced the removal effect of TC.

In our study, Ca2+ can affect TC removal by competing with TC for adsorption sites on HFO; thus influencing complexation interactions, especially outer-sphere complexation, affecting the ion exchange mechanism, weakening the electrostatic interaction, and Ca2+ could form a complex with TC, further affecting the adsorption of TC on HFO.

Comment 10: Line 296-298: The authors expressed that “the pseudo-second-order model better described the adsorption process and showed a higher correlation coefficient (>0.99). Thus, chemical interactions played a major role in the adsorption process”. The adsorption mechanisms cannot be directly assigned based on observing simple kinetic experiments or by fitting kinetic models. Please see the following research articles.

 Lima, E.C., Cestari, A.R., Adebayo, M.A., 2016. Comments on the paper: a critical review of the applicability of Avrami fractional kinetic equation in adsorption based water treatment studies. Desalination Water Treat. 57 (41), 19566e19571.

 Tran et al., 2017. Mistakes and inconsistencies regarding adsorption of contaminants from aqueous solutions: A critical review. Water Research 120 ,88-116.

Response 10:

Thanks very much for your comment.

The adsorption mechanism cannot be obtained based on the results of adsorption kinetis. Thus, FTIR and XPS was applied to characterize the HFO after TC adsorption. Meanwhile, the effect of pH and ion types was also investigated to explore the potential mechanism.

Thus, chemical interactions played a major role in the adsorption process [46]. The chemical interaction theoretically presented based on adsorption kinetics. The reference had been added in th revised manuscript.

Ho, Y. S.; McKay, G. Pseudo-second order model for sorption processes. Process Biochem. 1999, 34, 451-465.

Thus, the main mechanism of TC adsorption on HFO included electrostatic interactions, hydrogen bonding and complexation.

Many studies reporting that pseudo-second order kinetics fitted their data well did not conclude in general that the adsorption step actually controlledthe process. Sometimes, results were observed to be described satisfactorily by this ratelaw and, at the same time, diffusion (in the external layer and/or within the particle) wasfound to signi cantly contribute. In other cases, diffusion has been proposed as therate-controlling process.

Comment 11: What is the physical meaning of pseudo second order (and comparably of 1st order) and how this can be interpreting to the TC adsorption?

Response 11:

Thanks very much for your comment.

The overall kinetics of adsorbate adsorption is dependent on both the actual adsorption of the adsorbate onto the adsorption sites, defined by the corresponding kinetic and equilibrium constants, and its mass transfer towards these sites, characterized by the corresponding diffusion coefficient. Adsorption kinetic studies in liquid/solid systems are often conducted under batch conditions where the transient adsorbate concentration in the solution is fitted by a suitable kinetic model.

The physical meaning of the rate constant in the pseudo-second order model was interpreted on the basis of the transient adsorption in a model liquid/solid system, described by the analytical solution of the diffusion equation in spherical coordinates [1-3].

[1] Miyake, Y., Ishida, H., Tanaka, S., Kolev, S. D. (2013). Theoretical analysis of the pseudo-second order kinetic model of adsorption. Application to the adsorption of Ag (I) to mesoporous silica microspheres functionalized with thiol groups. Chemical engineering journal, 218, 350-357.

[2]Ho, Y. S., McKay, G. (1999). Pseudo-second order model for sorption processes. Process biochemistry, 34(5), 451-465.

[3] Simonin, J. P. (2016). On the comparison of pseudo-first order and pseudo-second order rate laws in the modeling of adsorption kinetics. Chemical Engineering Journal, 300, 254-263.

Comment 12: Line 306-307: The authors stated that “An increase in temperature promoted the collision between adsorbent and adsorbent, and enhanced the adsorption of TC on HFO”. What are the key arguments for your claim?

Response 12:

Thanks very much for your comment.

The sentence had been changed in the revised manuscript to better explain the influence of temperature on TC adsorption.

Temperature increasing can enhance the activation energy of TC molecules.

Temperature is a very important factor to be studied for adsorption. Temperature can affect adsorption by changing the sorption capacity and also influencing the molecules diffusion rate, thus increasing or decreasing the time for adsorption equilibrium.

As temperature rises, solution viscosity drops, which is favorable to the subsequent adsorption stages: external transfer, and diffusion of adsorbate within the adsorbent solid. This increase may be due to: i) the increase in TC molecules mobility, allowing it to penetrate the sample pores; ii) the increase in chemical interactions between the adsorbate and surface functionalities of the adsorbent; and iii) the change in chemical potentials, correlated with adsorbate species solubility.

Comment 13: Line 329 and Figure 8: What is Kd and how its value is calculated?

Response 13:

Thanks very much for your comment.

Some content had been added in the revised manuscript to present the calculation of Kd.

Where Kd is the apparent equilibrium constant; R is the ideal gas constant (8.314 J mol−1 K−1) and T is Kelvin temperature (K).

According to the method suggested by Khan and Singh [31], the sorption distribution coefficient Kd for the sorption reaction was determined from the slope of the plot ln(qe/Ce) against Ce at various temperatures and extrapolating to zero Ce. The values of ΔH and ΔS can be obtained from the slope and intercept of a plot of ln Kd against 1/T.

Khan, A. A.; Singh, R. P., Adsorption thermodynamics of carbofuran on Sn (IV) arsenosilicate in H+, Na+ and Ca2+ forms. Colloids Surf., 1987, 24, 33-42.

Comment 14: Table 4: How did you calculate the change in enthalpy (ΔH) and change in entropy (ΔS)?

Response 14:

Thanks very much for your comment.

Some content had been added in the revised manuscript to calculate in enthalpy (ΔH) and change in entropy (ΔS).

“Where Kd is the apparent equilibrium constant; R is the ideal gas constant (8.314 J mol−1 K−1) and T is Kelvin temperature (K).

According to the method suggested by Khan and Singh [34], the sorption distribution coefficient Kd for the sorption reaction was determined from the slope of the plot ln(qe/Ce) against Ce at various temperatures and extrapolating to zero Ce. The values of ΔH and ΔS can be obtained from the slope and intercept of a plot of ln Kd against 1/T.”

Comment 15:

Line 345-347: “the effect of ionic types and strengths showed that K+ had little influence on TC removal, suggesting that the complexation between HFO and TC was an inner-sphere complexation, and not an outer-sphere complexation? What are the key evidences to this conclusion?

Response 15:

Thanks very much for your comment.

According to the studies, ions types can influence the complexation between adsorbent and adsorbate.

According to previous study [1], ionic strength dependent adsorption indicates that ion exchange or outer-sphere complexation contributes to Pb adsorption on diatomite at pH< 7.0, the ionic strength independent adsorption suggests that inner-sphere complexation is the main adsorption mechanism of Pb(II) on diatomite at pH> 7.0

[1]Sheng, G., Wang, S., Hu, J., Lu, Y., Li, J., Dong, Y., & Wang, X. (2009). Adsorption of Pb (II) on diatomite as affected via aqueous solution chemistry and temperature. Colloids and Surfaces A: Physicochemical and Engineering Aspects, 339(1-3), 159-166.

Some content had been changed in the revised manuscript.

“The effect of ionic types and strengths showed that K+ had little influence on TC removal, suggesting that the complexation between HFO and TC may be an inner-sphere complexation, and not an outer-sphere complexation.”

Comment 16:

Line 348-349: The authors expressed that “The results of adsorption kinetics illustrated that chemisorption played a major role in the adsorption process”. Did you reach the conclusion solely based on the fitting of pseudo second order equation? If so, please state references from literature for this claim.

Response 16:

Thanks very much for your comment.

Reference had been added in the manuscript. The adsorption mechanism cannot be obtained based on the results of adsorption kinetis. Thus, FTIR and XPS was applied to characterize the HFO after TC adsorption. Meanwhile, the effect of pH and ion types was also investigated to explore the potential mechanism.

Comment 17: Please add to the introduction: Purpose of this study with respect to very high TC concentrations applied

Response 17:

Thanks very much for your comment.

Studies on monitoring of TC in the environment indicate that very low concentrations (μg/L to ng/L) present in treated waters and higher levels (100–500 mg/L) were detected in effluents of hospital and pharmaceutical manufacturing wastewaters [2-4].

First, the choice of the concentration in this study was due to a typical concentration in some environmental water. Second, this concentration could be detected by High Performance Liquid Chromatography (HPLC) after dilution.

Fu, Y., Peng, L., Zeng, Q., Yang, Y., Song, H., Shao, J., Shao, S., Gu, J. (2015). High efficient removal of tetracycline from solution by degradation and flocculation with nanoscale zerovalent iron. Chem. Eng. J. 2015, 270, 631-640 Jing, X. R., Wang, Y. Y., Liu, W. J., Wang, Y. K., Jiang, H. Enhanced adsorption performance of tetracycline in aqueous solutions by methanol-modified biochar. Chem. Eng. J. 2014, 248, 168-174. Ahmadi, M., Motlagh, H. R., Jaafarzadeh, N., Mostoufi, A., Saeedi, R., Barzegar, G., Jorfi, S. (2017). Enhanced photocatalytic degradation of tetracycline and real pharmaceutical wastewater using MWCNT/TiO2 nano-composite. J. Environ. Manage. 2017, 186, 55-63.

Comment 18: Conclusions must be extended considerably, not all claims from the abstract can be found here.

Response 18:

Thanks very much for your comment.

The conclusions had been rewritten in the revised manuscritp.

“The adsorption behavior of TC onto HFO could be affected by solution pH , ionic types and temperature. Pseudo-second-order adsorption kinetics and the Langmuir adsorption isotherm fitted the adsorption process well. Electrostatic interactions, hydrogen bonding and complexation were the main adsorption mechanism between TC and HFO. Therefore, the environmental behavior of TC could be affected by HFO and the transformation of TC affected by HFO in the aquatic environment should be studied in the future. ”

Reviewer 3 Report

In this work, the authors experimentally investigated the removal of tetracycline over prepared hydrous ferric oxide. Adsorption kinetics was discussed, and the mechanism was proposed based on the FTIR and XPS spectra. The manuscript is well written, and I found it publishable after addressing the minor issues below.

On Page 4, line 133 and 134, in the equations of the kinetic models, is the “x” supposed to be “t”? On Page 7, line 216, how was the optimal condition defined? When the dosage was 1.0, neither the removal rate nor the adsorption capacity was at the maximum. Before running the adsorption experiments for kinetics, the authors need to demonstrate the significance of the mass transfer resistance under the chosen conditions. On Page 8, line 298, based on the data fitting, the authors suggested chemical interaction play a role. Again, it is difficult to jump to this interpretation before excluding the significance of mass transfer resistance.

        Similarly, on Page 9, line 307, the authors attributed the reason of higher adsorption capacity at higher temperatures to a better adsorption reaction. However, it could be due to an improved mass transfer at higher temperatures.  

Author Response

Reply to the Comments of Reviewer 3

Comment 1: On Page 4, line 133 and 134, in the equations of the kinetic models, is the “x” supposed to be “t”? On Page 7, line 216, how was the optimal condition defined? When the dosage was 1.0, neither the removal rate nor the adsorption capacity was at the maximum. Before running the adsorption experiments for kinetics, the authors need to demonstrate the significance of the mass transfer resistance under the chosen conditions. On Page 8, line 298, based on the data fitting, the authors suggested chemical interaction play a role. Again, it is difficult to jump to this interpretation before excluding the significance of mass transfer resistance.

Response 1:

Thanks very much for your comment.

The “x” had been replaced by “t” in the revised manuscript.

Based on the consideration of removal effect and cost, the optimum HFO dosage was chose as 1.0 g·L-1.

Some content had been added in the revised manuscript to demonstrate the significance of the mass transfer resistance. In fact, the adsorption mechanism cannot be obtained based on the results of adsorption kinetics. Thus, FTIR and XPS were applied to characterize the HFO after TC adsorption. Meanwhile, the effect of pH and ion types

were also investigated to explore the potential mechanism.

Thus, chemical interactions played a major role in the adsorption process [46]. The chemical interaction theoretically presented based on adsorption kinetics.

Ho, Y. S.; McKay, G. Pseudo-second order model for sorption processes. Process Biochem. 1999, 34, 451-465.

Absolutely, the significance of mass transfer resistance is important factor for the adsorption process, especially at the initial stage. Furthremore, the chemical interaction was confirmed by FTIR and XPS analysis.

Comment 2: Similarly, on Page 9, line 307, the authors attributed the reason of higher adsorption capacity at higher temperatures to a better adsorption reaction. However, it could be due to an improved mass transfer at higher temperatures.  

Response 2:

Thanks very much for your comment.

The content had been changed in the revised manuscript.

“Increasing temperature can affect adsorption by changing the sorption capacity and also influencing the molecules diffusion rate, thus increasing or decreasing the time for adsorption equilibrium and then enhanced the adsorption of TC on HFO.”

Reviewer 4 Report

line 53/54: sentence?

line 122: HCl is missing behind 0.1 mol L-1

line 150: value of the ideal gas constant is wrong (R= 8.314 J/mol*K)

Often words are split wrongly: e.g. 533.24 eV in line 180 or "amphoteric molecule.......and may"...in line 233

line 233/234: different form of ions: all in singular or in plural

Table 2: T/°C ? and R22 ?

line 307: "collision between adsorptive and adsorbent"

line 334/335: ΔH < 40 kJ/mol means Physisorption; ΔH > 50 kJ/mol means Chemisorption that means that the conclusion from the ΔH value is wrong.

Author Response

Reply to the Comments of Reviewer 4

Comment 1: line 53/54: sentence?

Response 1:

Thanks very much for your comment.

The setence had been supplemented in the revised manuscript.

“However, the presence of competing cations, the influence of temperature and potential mechanism should be studied further.”

Comment 2: line 122: HCl is missing behind 0.1 mol L-1

Response 2:

Thanks very much for your comment.

HCl had been added in the revised manucript.

Comment 3: line 150: value of the ideal gas constant is wrong (R= 8.314 J/mol*K)

Response 3:

Thanks very much for your comment.

The value of ideal gas constant had been revised manuscript.

Comment 4: Often words are split wrongly: e.g. 533.24 eV in line 180 or "amphoteric molecule.......and may"...in line 233

Often words are split wrongly: e.g. 533.24 eV in line 180 or "amphoteric molecule.......and may"...in line 233

Response 4:

Thanks very much for your comment.

The wrongly split between words had been revised in the manuscript.

Comment 5: line 233/234: different form of ions: all in singular or in plural

Response 5:

Thanks very much for your comment.

The form of ions had been kept consistence.

“Tetracycline (symbolized as H2TC) is an amphoteric molecule with multiple ionizable functional groups andmay exist as a cations (H3TC+, pH<3.3), zwitterions (H2TC0, 3.3<pH<7.7), or negatively charged ions (HTC−, 7.7<pH<9.7; TC2−, pH>9.7) at different pH values [38].”

Comment 6: Table 2: T/°C ?and R22 ?

Response 6:

Thanks very much for your comment.

The description had been changed in the revised table.

Comment 7: line 307: "collision between adsorptive and adsorbent"

Response 7:

Thanks very much for your comment.

The content had been changed in the revised manuscript to better explain the influence of temperature.

“Increasing temperature can affect adsorption by changing the sorption capacity and also influencing the molecules diffusion rate, thus increasing or decreasing the time for adsorption equilibrium and then enhanced the adsorption of TC on HFO. ”

Comment 8: line 334/335: ΔH < 40 kJ/mol means Physisorption; ΔH > 50 kJ/mol means Chemisorption that means that the conclusion from the ΔH value is wrong.

Response 8:

Thanks very much for your comment.

The content had been changed in the revised manuscript. Both physical sorption and chemical sorption play a role during the adsorption process.

The ΔH>0 indicated that the adsorption process was endothermic. In this study, ΔH was 10.1431 kJ mol−1 which was less than 80 kJ mol−1 [55], indicating that physical binding also plays an important role during the adsorption process.

Chowdhury, S.; Mishra, R.; Saha, P.; Kushwaha, P. Adsorption thermodynamics, kinetics and isosteric heat of adsorption of malachite green onto chemically modified rice husk. Desalination2011, 265, 159-168.

Round 2

Reviewer 1 Report

Authors have improved the manuscript, but I have two suggestions

In question 4, the authors said they rounded the figures in table 1, however, in the revised version there are the same data; 1629.01 cm-1, etc..

Regarding XPS Fe2p spectra, there are two peaks associated with Fe species, Can authors add the description of the Fe species in the HFO?.

Author Response

Journal: International Journal of Environmental Research and Public Health

Manuscript Number: ijerph-619304

Title: Removal of tetracycline by hydrous ferric oxide: adsorption kinetics, isotherms and mechanism

Authors: Ji Zang, Tiantian Wu, Huihui Song, Nan Zhou, Shisuo Fan, Zhengxin Xie and Jun Tang*

The authors appreciate the kind comments from the Editor and the Reviewers, and respond as follows.

The changes during revision had been appeared in a blue color in the “revised manuscript”.

Reply to the Comments of Reviewer 1

Comment 1: In question 4, the authors said they rounded the figures in table 1, however, in the revised version there are the same data; 1629.01 cm-1, etc..

Response 1:

Thanks very much for your comment.

The figures had been rounded in the table 1.

The wavenumbers in table 1 was consistence with the peaks in figure 1(b).

“The FTIR spectra of HFO, TC, and TC adsorbed onto HFO are displayed in Figure 1(b) and specific functional groups are listed in Table 1. The main functional groups of HFO included –OH (3405 cm-1, 1629 cm-1), CH3/COO (1477 cm-1, 1348 cm-1) and Fe-O (444 cm-1). After TC was adsorbed on HFO, some new peaks appeared, involving amino groups (1535 cm-1) and C-OH stretching (1224 cm-1), suggesting that TC had been adsorbed on the surface of HFO.”

“In addition, following TC adsorption, the peaks at 1629 cm-1, 1477 cm-1 and 444 cm-1 shifted to 1617 cm-1 , 1458 cm-1 and 436 cm-1, respectively, indicating that Fe-O, CH3/COO and -OH in HFO participated in the removal of TC in solution. The main mechanism involved electrostatic interactions and complexation. Mudunkotuwa et al. [37] found that the ATR-FTIR spectra of α-Fe2O3-adsorbed humic acid showed two new bands at 1348 and 1470 cm−1 during the initial time points, which corresponded to carboxylate groups strongly adsorbed to the surface iron atoms.

Comment 2: Regarding XPS Fe2p spectra, there are two peaks associated with Fe species, Can authors add the description of the Fe species in the HFO?.

Response 2:

Thanks very much for your comment.

Some content had been added in the revised manuscript.

“The Fe 2p XPS spectrum of HFO was separated into four peaks at 710.50, 712.23, 718.51 and 724.63 eV, corresponding to Fe 2p3/2, Fe 2p3/2, Fe 2p1/2 and Fe 2p1/2, respectively. The Fe species in HFO refers to Fe2O3, FeO, FeOOH, which was consistent with the analysis of XRD.”

Furthermore, the whole manuscript had been carefully checked and some mistakes had been revised in the “revised manuscript”. In the end, thanks very much for the precious comments from the Editor and Reviewers to make a better manuscipt.

Best wishes!

Reviewer 2 Report

See attached file

Author Response

Journal: International Journal of Environmental Research and Public Health

Manuscript Number: ijerph-619304

Title: Removal of tetracycline by hydrous ferric oxide: adsorption kinetics, isotherms and mechanism

Authors: Ji Zang, Tiantian Wu, Huihui Song, Nan Zhou, Shisuo Fan, Zhengxin Xie and Jun Tang*

The authors appreciate the kind comments from the Editor and the Reviewers, and respond as follows.

The changes during revision had been appeared in a blue color in the “revised manuscript”.

Reply to the Comments of Reviewer 2

Comment 1:

See attached file

Response 1:

Thanks very much for your comment.

We cannot find the comment in the attached file.

We had responded the comment in the previous attached file.

Furthermore, the whole manuscript had been carefully checked and some mistakes had been revised in the “revised manuscript”. In the end, thanks very much for the precious comments from the Editor and Reviewers to make a better manuscipt.

Best wishes!
